# Eating Habits and Physical Activity of the Spanish Population during the COVID-19 Pandemic Period

**DOI:** 10.3390/nu12092826

**Published:** 2020-09-15

**Authors:** Eduardo Sánchez-Sánchez, Guillermo Ramírez-Vargas, Ylenia Avellaneda-López, J. Ignacio Orellana-Pecino, Esperanza García-Marín, Jara Díaz-Jimenez

**Affiliations:** 1Punta de Europa Hospital, Algeciras, 11207 Cádiz, Spain; guiram1992@gmail.com (G.R.-V.); ain3ly@gmail.com (Y.A.-L.); espe_garciamarin@hotmail.com (E.G.-M.); 2Faculty of Education Sciences, University of Cádiz, 11519 Puerto Real, Spain; joseignacio0107@gmail.com (J.I.O.-P.); luna_nueva17@hotmail.com (J.D.-J.)

**Keywords:** COVID-19, Mediterranean Diet, diet, physical activity, confinement

## Abstract

Due to the pandemic situation caused by the COVID-19 infection, some governments have implemented house confinement measures. The objective of our study is to learn the dietary patterns, consumption, and physical activity of the Spanish population before and during the period of confinement by COVID-19. A cross-sectional descriptive study based on a questionnaire during May 2020, coinciding with the period of confinement and the step forward into Phase 1, is carried out. During confinement, the adherence to the Mediterranean Diet increases (8.0% versus 4.7%; *p* < 0.001). No socio-demographic variables show statistical significance (*p* < 0.05) regarding good adherence to the Mediterranean Diet (MD) before and during confinement. During confinement, consumption of homemade baking shows a higher increase (0.28% versus 4.60%; *p* = 0.004). During confinement, the number of subjects that practice exercise decreases (29.4% versus 28.8%; *p* = 0.004), as well as the time spent exercising (more than an hour, 26.6% versus 14.7%, *p* = 0.001). Mediterranean Diet adherence slightly increases during confinement, although consumption of ‘unhealthy’ food also increases. Moreover, the number of subjects that practice physical activity, as well as the time spent on it weekly, decreases.

## 1. Introduction

A number of cases of atypical pneumonia caused by a new virus with a rapid spread, infectivity, and mortality in human beings are noticed in Wuhan, China, in December 2019. Subsequently, the World Health Organisation (WHO) temporarily names this new virus as the new coronavirus 2019 (2019-nCoV), and its disease as COVID-19. The WHO has defined this situation as a world public health emergency. Regarding clinical manifestations, the most severe is the severe acute respiratory syndrome (SARS), which is why the virus has been renamed as SARS CoV-2 [1,2,3].

This pandemic has led to restrictive measures and house confinement by governments of multiple countries, with the aim of reducing the number of cases and its spread within the population [4].

Confinement periods imply daily routine and lifestyle changes for the population. Some studies have linked the period of confinement with an increase in negative psychological effects, such as stress and anxiety [5]. The confinement itself, and its negative psychological effects, may lead to inappropriate conduct like physical inactivity, increases in a sedentary lifestyle, and changes in dietary patterns toward the consumption of ‘unhealthy’ food and beverages. This conduct might increase the risk of developing diseases such as obesity, diabetes, cardiovascular disease, cancer and more [6,7], and these diseases are mortality risk factors for COVID-19 [8].

It is advisable, therefore, to maintain physical activity in periods of confinement and to follow a healthy diet. This diet should be rich in whole grains, vegetables, fruits, legumes, and healthy fats. The Mediterranean Diet (MD) follows this eating pattern and it has shown different benefits like reduction in ischemic cardiopathology, an improved lipid profile, lower blood pressure, insulin resistance, and less risk of cancer and strokes [9,10,11]. Dietary habits, food consumption patterns, and physical activity knowledge may help national governments make more effective and efficient health policies during a potential new period of confinement. Additionally, it is important for the population to use the Information and Communication Technologies (ICTs).

The objective of our study is to evaluate the eating habits, consumption, and physical activity of the Spanish population before and after a COVID-19 confinement period.

## 2. Materials and Methods

### 2.1. Selection of Participants and Study Design

A cross-sectional descriptive study based on a self-administered questionnaire was carried out. This was a non-probabilistic sample used for convenience as this questionnaire was directed to the whole Spanish population above sixteen years old.

To calculate the sample size, the data published by The National Statistical Institute (INE) on the 1st of July, 2019 were taken as reference, where it reflected that the Spanish population was composed of 47,100,936 inhabitants. The calculation of the sample size was carried out with a 95% confidence level and a 3% of precision, since the expected proportion of the change in population was unknown, a 0.5 proportion was selected. The sample size was 385 subjects.

The study followed the international ethical recommendations contained in The Declaration of Helsinki. Participation in the study was completely free and voluntary. All of the questionnaires were anonymous, and any personal identification data that was requested (name, ID) complied with the provisions of the Organic Law 15/1999, of December 13, Protection of Character Data Personal and Organic Law 3/2018, of December 5, for the protection of Personal Data and guarantee of digital rights.

### 2.2. Instruments and Variables

The questionnaire was divided into 3 sections. The first one included socio-demographic variables such as gender, age, autonomous region and place of residence, and professional situation. Anthropometric variables were taken in the second section, such as weight, height and Body Mass Index (BMI), as well as variables regarding food and nutrition, like adherence to the Mediterranean Diet, higher food consumption and participation in food preparation. The third, and last section, collected variables related to physical activity such as physical exercise, dedicated time, type of exercise, and information research to perform the exercise.

Regarding the Mediterranean Diet, adherence evaluation used the questionnaire from the Prevention with Mediterranean Diet group (PREDIMED) [12]. This questionnaire was validated for the Spanish population and consisted of 14 items. When the answer to the item followed the pattern of the Mediterranean Diet (MD) it scored 1 point. When the result obtained from the sum of the items was <9, subjects were classified as low adherence and if it was ≥9, subjects were considered to maintain a high adherence.

There existed few validated questionnaires for the physical activity assessment in the context of the investigation, but none of them were reproducible and, therefore, applicable during the confinement, consequently questions related to physical activity were designed based on opinions of physical activity experts.

A pilot study was carried out among 20 subjects to verify the effectiveness of the questionnaire, to learn whether it provided the necessary information, and if we needed to modify any of the questions. After this pilot study some questions were modified due to a lack of understanding which could lead to an interpretation error.

### 2.3. Data Collection

Concerning the dissemination of the questionnaire, new communication technology using the Google questionnaire platform was employed, and dissemination used social platforms on Twitter, Facebook, WhatsApp, and Instagram. The questionnaire was administered during May 2020, coinciding with the confinement period and the step forward into Phase 1.

### 2.4. Statistical Analysis

The data obtained from the variables were represented in a descriptive way. The qualitative variables were represented by frequency and percentage, and the quantitative variables were expressed by the mean and standard deviation or dispersion. Subsequently, and using the McNemar test, it was studied whether there were significant differences between gender and diet adherence, as well as the answers to each questionnaire item and the adherence to the MD during each period (before and after the confinement), accepting a confidence level of 95%.

Furthermore, a generalised linear regression model with all the variables and the adherence to MD during both periods of the study was conducted.

## 3. Results

### 3.1. Participant Characteristics

A total of 1073 answers to the questionnaire were obtained, of which eight were eliminated due to discrepancies within the responses related to age, weight and/or height. Considering the total of the sample, 72.8% were females and 27.2% were males. The average age of the sample was 38.7 ± 12.4 years, although, if we present the data by age ranges, the most prevalent were the subjects with ages between 26 and 40 years; being ≥71 years old was the least represented group in the sample.

BMI was calculated before confinement with 57.3% of those surveyed presenting values indicating normal weight. Any level of overweight or obesity was 39.8%. There were statistically significant differences within gender (*p* < 0.001) as the percentage of men with any level of overweight was higher than women (41.7% versus 22.9%). The next question related to weight regarded a lack of weight gain during confinement, which was reported at 47.3%. Concerning the subjects that confirmed a weight gain, 37.3% gained between 1 and 3 kg. There were not any statistical differences.

Among other evaluated aspects was professional situation, concerning on-site work, but not during the state of the initial pandemic, and telework, which were the most prevalent (23.3 and 22.0%, respectively). Considering gender, the number of unemployed women was higher than men (15.0% versus 9.0%).

Eighty-four point three percent of subjects lived in an urban area, 39.3% lived with their partners and children, and the percentage that lived with their parents and children was 1.3%. Additionally, 49.4% responded that they always cooked, this percentage being higher in women than men (55.2% versus 33.8) (Table 1).

### 3.2. Eating Habits and Adherence to MD

After analysing the answers obtained in the PREDIMED questionnaire, previous to confinement, 95.3% of the subjects presented a low adherence to the MD and 4.7% a high adherence. However, during confinement, 92.0% of subjects surveyed presented a low adherence compared to 8.0% that presented a high adherence. There were no statistically significant differences between genders during either period of time, although there were significant differences before and after confinement (*p* < 0.001), as subjects with a low adherence decreased (Table 2).

After the calculation of the Odds Ratio, any socio-demographic variable showing a statistical significance (*p* > 0.05) related to a good adherence to the MD before confinement. Following the confinement, those respondents living in Aragón were more likely to have a good adherence to the MD (Odds ratio (OR) = 4.39; Confidence interval (CI): 0.92–15.89; *p* = 0.0034). There were no differences between the different socio-demographic variables and improved observance of the MD during confinement.

Aside from evaluating MD adherence, each answer to each item of the PREDIMED questionnaire was studied. Answers providing 1 point per item were noted as it indicated a higher adherence to the MD. The items that presented a higher percentage, before confinement, were the use of olive oil for cooking, the consumption of butter, margarine, or cream, and the consumption of red meat, hamburgers, sausages, or cold meats, and the intake of carbonated beverages, at 97.9%, 96.5%, 87.1% and 86.5%, respectively. These were the same items that had a higher adherence, during confinement, although values were lower than those previously mentioned (97.6%, 91.7%, 82.7%, and 81.7%, respectively). Items with less adherence, before confinement, were the consumption of wine, fish or seafood, and legumes, at 3.1%, 23.9% and 25.4%, respectively. During confinement, the adherence to these items increased (8.1%, 28.4%, and 31.5%, respectively). There were significant statistical differences, between both periods, in all the items except for the usage of olive oil for cooking (*p* = 0.375) (Table 3).

Prior to the confinement the most consumed food and beverages by the studied subjects were chicken meat, turkey, or rabbit (20.47%), vegetables (14.46%) and pasta or rice (11.92%). The food less consumed, in this period, were homemade desserts and pastries (0.28%) and alcoholic drinks (1.03%). During confinement, the consumption of homemade desserts and pastries increased to 4.60%, the only one that presented a further increase (Table 4).

### 3.3. Physical Activity

When asking the subjects whether they practiced exercise before confinement, 35.4% answered that they did exercise 1–3 times per week, with a lower percentage during confinement (32.3%). The number of those surveyed that practiced exercise six or more times per week increased during confinement (7.9% versus 14.5%). Statistically significant differences existed in both genders and periods of time evaluated. Regarding the time spent for exercise, 8.3% dedicated between 10 and 30 min before confinement, increasing this percentage after confinement (21.4%). The number of surveyed subjects that responded who did more than an hour exercise per session decreased after confinement (26.6% versus 14.7%). There were statistically significant differences between gender and periods of time (Table 5).

Additionally, results showed that of the subjects that practiced physical activity during confinement, 36.16% searched for information to carry out physical activity using YouTube, while other social networks were less prevalent (7.74%) (*p* < 0.001) (Figure 1).

## 4. Discussion

Following a balanced and healthy diet, along with physical activity, plays an essential role in the maintenance of health in a population. These habits have gained a special relevance during the period of confinement caused by COVID-19, as the alteration of the diet and physical activity can yield diseases like obesity, diabetes; which are risk mortality factors in patients with COVID-19 [13].

According to reported data from the Spanish Agency of Food Security and Nutrition, 53% of the Spanish adult population present some level of overweight or obesity [14]. Prior to the confinement, in our sample, the prevalence of men that were overweight or obese was 52%, while this was less prevalent in women (35.2%). The number of obese subjects decreased during confinement, increasing the number of subjects who were overweight, which indicates a weight loss as height remained constant. This result is opposite to the results obtained by Pellegrini and colleagues in 2020, as those authors found a significant weight gain in the obese population. Consequently, it might be concluded that the percentage of obese subjects remained constant or increased [15]. Considering our subjects, 52.7% gained weight during confinement, which is a bigger percentage than the one reported by Sidor and Rzymski in Poland (29.9%) [16].The difference might be due to Poland having a confinement of 20–26 days (the 25th–31st of March to April the 20th), while in Spain it was 63 days (March 14th to May 2nd, first phase).

Home cooking makes a population have an active role in their dietary habits and it could influence a healthy diet follow-up [17]. This is not strictly correct, as cooking is the action of preparing food, but not the election of the food or the cooking process (roast, boiled, fried, ...). Considering our sample, 49.4% of respondents always cooked, which was more prevalent in women (55.2%) than in men (33.8%). Taking subjects that cook as a reference, without measuring the frequency of this action, 93.6% of the subjects cooked, with this value much more superior to the 44% shown in the COVIDiet study [18].

Our data show that MD adherence increased during confinement compared to prior to confinement. This increase is higher in women than men (8.5% versus 6.6%, respectively). Prior to confinement, 9.5 out of 10 surveyed subjects had a low adherence to the MD. These data are better in comparison to those reported in the study by Zaragoza–Martí et al., in 2015 for the Spanish population (51.7%) [19]. During a study performed in Italy, during confinement, 21.7% of people had a low adherence (numeric value 0–5), and 63.1% of people had a medium adherence (numeric value 6–9) to the MD. Although the cutting values are different from those taken as reference in our study, we can conclude that the majority of the Spanish population did not present a high adherence to the MD, similar data as that found in our study [20].

Our results show that none of the studied variables (age, gender, BMI, place of residence, professional situation, family members, cooking) had a relation to a high adherence to the MD before confinement, and only the place of residence (Aragon) was related with a greater probability of presenting a higher adherence to MD during confinement. During the COVIDiet study published in 2020, age, educational level, and region were connected to MD adherence before and after confinement, regarding subjects with an age range between 21–50 years old, or with a superior level of education, or who lived in the north, all of which presented a higher adherence to the MD [18]. Also, other studies showed that adults over 40 years old, living with children, and unemployed had a higher risk for unhealthy food consumption [21]. Those differences might be due to the number of participants and the study’s chosen questionnaire (PREDIMED versus MetDiet).

Prior to and during confinement, the PREDIMED questionnaire items which presented a higher adherence in our population were: consumption of olive oil for cooking (97.9% versus 97.6%); use of butter, margarine, or cream (96.5% versus 91.7%); consumption of red meat, hamburgers, sausages, or cold meats (87.1% versus 82.7%); and consumption of carbonated beverages and/or sweetened drinks (86.5% versus 81.7%). Those which presented a lower adherence were: consumption of wine (3.1% versus 8.1%); consumption of fish and/or seafood (23.9% versus 28.4%), legumes (25.4% versus 31.5%), and fruits (27.5% versus 35.7%). There were statistical differences within all the items before and during confinement except for olive oil for cooking and vegetable consumption. These data are different than those reported by Di Renzo et al., [20], with the exception of the consumption of olive oil for cooking (95.8%). Regarding the previously mentioned study, adherence to the MD is associated with a higher intake of fruits, vegetables, nuts, legumes, and fish. 

Regarding food consumption changes, data show that the consumption of alcoholic drinks, confectionaries, nuts, homemade desserts and snacks, and jelly beans increased during confinement, although this increase was lower than that reported in other studies such as the one by Scamozzino and Visioli in 2020, which obtained a 10.1% increase in the consumption of alcohol drinks versus 1.79% in our study. Furthermore, in the previously mentioned study, 21.2% of respondents increased fruit and vegetable consumption, being this increase lower than in our sample (0.37% fruits and 0.19% vegetables). This might be due to the difference between samples or to the initial consumption of these products [22]. Moreover, in Spain, the purchase of snacks, baking flour, and bread suffered a substantial increase. During confinement, chicken, turkey and/or rabbit meat, and pasta and rice consumption decreased.

Physical activity plays an essential role in the achievement of the beneficial effects of the food in health, as they are like a tandem that always must be together: healthy diet and physical exercise. Concerning our studied sample, 28.8% did not practice any physical activity before confinement, which slightly increased to 29.4% during confinement. This lack of physical activity is more remarkable in women than men (35.4% versus 11.4% prior to confinement and 31.5% versus 23.8% during confinement). Our data are closer to those reported by Zaragoza–Martí et al., in 2015 [19] where 31.7% and 37.4% of the subjects did not exercise before and during confinement, respectively. The increase in subjects that did not practice any physical exercise was lower in our study. Our outcomes were higher than those reported by other studies, regarding the number of subjects that practiced physical activity during confinement (70.6% versus 59.6%) [18].

Regarding weekly sessions of physical activity, subjects who practiced between 1–3 sessions per week were the most prevalent in our sample, slightly decreasing during confinement, as well as subjects that practiced 4–5 times per week. Subjects that practiced ≥6 sessions per week increased. This rise in the number of sessions during confinement also was reflected in other studies [19]. Furthermore, the number of participants that spent 10–30 min in physical practice during confinement increased (8.3% versus 21.4%), while decreasing the number of those who dedicated more than an hour (26.6% versus 14.7%). The increase in the number of sessions offset the time reduction, as the suggested objective for people who remained at home due to COVID-19 confinement could be achieved (150 min per week) [23].

Seen in a recent study, it was proved that confinement led to a greater usage of ICTs (15%) [24]. Our results show that 43.9% of the subjects used ICTs for exercise research, videos and/or advice about physical activity. Videos about Yoga, Pilates, or Zumba were the most searched online.

Considering the strengths or contributions of our study, the main one is the examination of dietary patterns, consumption, and physical activity, all together, during an atypical period such as the domiciliary confinement. Dietary habits and physical activity can be protective factors against weight gain during confinement, which would avoid the increase in obesity, diabetes, cardiovascular diseases, etc. [25].

Recent figures make us think how we could get back to this situation again, due to the number of people who become infected by the SARS CoV-2 virus, and the knowledge of these patterns may help us to propose strategies to fight physical inactivity and unbalanced diets, thus reducing the risk of disease as a result of them. It should not be forgotten, the promotion of teleworking and the follow-up of online lessons from students is attached to the increase in time sitting.

One of the limitations found in our study, that could give rise to a selection bias, is a lack of similarity in the representation of each geographical area. Additionally, there exists a larger percentage of women than men. The usage of a prepared questionnaire by a research team, due to the lack of a validated questionnaire, may prompt a bias, as questions might be directed and relevant information about other physical exercise factors may be lost. Due to COVID-19 restriction, there were difficulties in performing person-to-person questionnaires or interviews, requiring use of an online questionnaire where those subjects who do not receive that kind of information or who do not have internet access are excluded.

## 5. Conclusions

Mediterranean diet adherence slightly increased during confinement, however the consumption of ‘unhealthy’ food such as alcoholic beverages, snacks and sweets, confectionaries, also increased. Additionally, the number of subjects that practiced physical activity decreased, as well as the time spent active weekly, although there was an increase in the number of sessions/week.

The knowledge of eating habits and the practice of physical activity in the Spanish population during confinement should guide government and/or academic agencies to propose strategies that could encourage a balanced and healthy diet (MD) and physical activity practice during a new period of confinement or restrictions caused by COVID-19.

## Figures and Tables

**Figure 1 nutrients-12-02826-f001:**
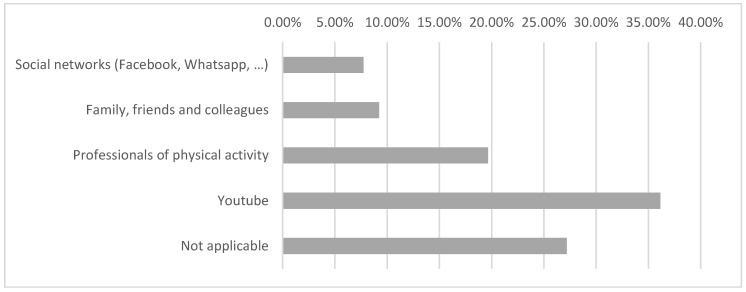
Search for information for physical exercise.

**Table 1 nutrients-12-02826-t001:** Socio-demographic, anthropometric and socio-economic variables.

Variables	*n*	%	Gender		*p*-Value
			Female; *n* (%)	Male; *n* (%)	
			775 (72.8)	290 (27.2)	
**Age (intervals):**					*p* = 0.185
16 to 25 years.	191	17.9	137 (17.7)	54 (18.6)	
26 to 40 years.	415	39	299 (38.6)	116 (40.0)	
41 to 55 years.	350	32.9	266 (34.3)	84 (29.0)	
56 to 70 years.	105	9.8	69 (8.9)	36 (12.4)	
≥71 years.	4	0.4	4 (0.5)	0 (0.0)	
**BMI category:**					*p* < 0.001
Underweight.	31	2.9	27 (3.5)	4 (1.4)	
Normal weight.	510	57.3	475 (61.3)	135 (46.6)	
Overweight level I.	135	12.7	77 (9.9)	58 (20.0)	
Overweight level II.	164	15.4	101 (13.0)	63 (21.7)	
Obesity type I.	93	8.7	70 (9.0)	23 (7.9)	
Obesity type II.	23	2.2	17 (2.2)	6 (2.1)	
Obesity type III:	9	0.8	8 (1.0)	1 (0.3)	
**Weight gain during confinement.**					*p* = 0.305
Between 1 and 3 kg.	355	37.3	259 (33.4)	96 (33.1)	
Between 4 and 7 kg.	74	6.9	48 (6.2)	26 (9.0)	
Between 8 and 11 kg.	5	0.5	3 (0.4)	2 (0.7)	
>12 kg	3	0.4	2 (0.3)	1 (0.3)	
Yes, but I do not know how many kg.	124	11.6	99 (12.8)	25 (8.6)	
No	504	47.3	364 (47.0)	140 (48.3)	
**Professional status:**					*p* = 0.002
Self-employed, open business.	8	0.7	3 (0.4)	5 (1.7)	
Self-employed, close business.	23	2.2	17 (2.2)	6 (2.1)	
Unemployed.	142	13.3	116 (15.0)	26 (9.0)	
Layoff.	108	10.1	71 (9.2)	37 (12.8)	
Student.	161	15.1	119 (15.4)	42 (14.5)	
On-site work during the whole period.	102	9.6	63 (8.1)	39 (13.4)	
On-site work part of the period.	248	23.3	180 (23.2)	68 (23.4)	
Telework.	234	22	181 (23.4)	53 (18.3)	
Retired.	39	3.7	25 (3.2)	14 (4.8)	
**Place of residence:**					*p* = 0.001
Rural area.	167	15.7	138 (17.8)	29 (10.0)	
Urban area.	898	84.3	637 (82.2)	261 (90.0)	
**Number of people in the residence:**					*p* = 0.010
Alone.	103	9.7	77 (9.9)	26 (9.0)	
Couple.	223	20.9	152 (19.6)	71 (24.5)	
With children and without partner.	54	5.1	50 (6.5)	4 (1.4)	
Couple and children.	419	39.3	309 (39.9)	110 (37.9)	
Parents.	227	21.3	156 (20.1)	71 (24.5)	
Couple, children, and parents.	25	2.4	19 (2.5)	6 (2.1)	
Parents and children.	14	1.3	12 (1.5)	2 (0.7)	
**Home cooking.**					*p* < 0.001
Always	526	49.4	428 (55.2)	98 (33.8)	
Several occasions.	258	24	183 (23.6)	73 (25.2)	
Few occasions.	215	20.2	128 (16.5)	87 (30.0)	
Do not cook.	68	6.4	36 (4.6)	32 (11.0)	

BMI: Body mass index.

**Table 2 nutrients-12-02826-t002:** Adherence to MD before and during confinement outcomes.

PREDIMED Outcome	Before Confinement			During Confinement		
	Female	Male	Total	Female	Male	Total
	*n* (%)	*n* (%)	*n* (%)	*n* (%)	*n* (%)	*n* (%)
Low Adherence	740 (95.5)	275 (94.8)	1015 (95.3)	709 (91.5)	279 (93.4)	980 (92.0)
High Adherence	35 (4.5)	15 (5.2)	50 (4.7)	66 (8.5)	19 (6.6)	85 (8.0)
	X^2^ = 0.20 *; *p* = 0.652			X^2^ = 1.10 *; *p* = 0.292		
	X^2^ = 274.76 *; *p* < 0.001					

MD: Mediterranean Diet; PREDIMED: questionnaire from the Prevention with Mediterranean Diet group, * McNemar–Bowker test. X^2^: chi square; *p*: *p*-value.

**Table 3 nutrients-12-02826-t003:** Distribution of responses in terms of MD adherence.

Questions/Answers to Questionnaire	Before Confinement			During Confinement			*p*
	Female	Male	Total	Female	Male	Total	
**Do you use olive oil for cooking?**							0.375
Yes	97.80%	98.30%	97.90%	97.50%	97.90%	97.60%	
No	2.20%	1.70%	2.10%	2.50%	2.10%	2.40%	
**How much olive oil do you consume daily (including to fry, to cook, salads)?**							<0.001
3 spoons or less.	52.40%	54.10%	52.90%	45.20%	52.40%	47.10%	
4 spoons or more.	47.60%	45.90%	47.10%	54.80%	47.60%	52.90%	
**How many portions of vegetables do you consume every day? (Garnishes and accompaniments would be ½ portion, 1 portion is equal to 200 g)**							0.032
1 or less.	27.10%	40.30%	30.70%	24.90%	37.20%	28.30%	
2 or more, none of them in salad or raw.	17.40%	17.90%	17.60%	19.00%	22.80%	20.00%	
2 or more, some of them in salad or raw.	55.50%	41.70%	51.70%	56.10%	40.00%	51.70%	
**How many pieces of fruit, including fresh juice, do you consume every day?**							<0.001
2 or less per day.	72.90%	71.40%	72.50%	63.70%	65.90%	64.30%	
3 or more per day.	27.10%	28.60%	27.50%	36.30%	34.10%	35.70%	
**How many portions of red meat, hamburgers, sausages, or cold meat do you consume every day? (portion 100–150 g)**							<0.001
1 or less per day.	87.90%	85.20%	87.10%	83.40%	81.00%	82.70%	
2 or more per day.	12.10%	14.80%	12.90%	16.60%	19.00%	17.30%	
**How many portions of butter, margarine or cream do you consume every day? Individual portion = 2 g.**							<0.001
1 or less per day.	96.10%	97.60%	96.50%	91.00%	93.80%	91.70%	
2 or more per day.	3.90%	2.40%	3.50%	9.00%	6.20%	8.20%	
**How many carbonated and/or sugary beverages (Soft drinks, Cola, tonic…) do you consume every day?**							<0.001
1 or less per day.	87.40%	84.10%	86.50%	81.80%	81.40%	81.70%	
2 or more per day.	12.60%	15.90%	13.50%	18.20%	18.60%	18.30%	
**Do you drink wine? How much do you consume per week?**							<0.001
Less than 7 times per week.	97.80%	94.50%	96.90%	92.40%	90.70%	91.90%	
7 or more times per week.	2.20%	5.50%	3.10%	7.60%	9.30%	8.10%	
**How many portions of legumes do you consume per week? (1 dish or portion is 150 g).**							<0.001
2 or less portions per week.	75.00%	73.40%	74.50%	68.30%	69.00%	68.40%	
3 or more portions per week.	25.00%	26.60%	25.40%	31.70%	31.00%	31.50%	
**How many portions of fish/seafood do you consume per week? (1 dish, piece, or portion = 100–150 g of fish or 4–5 pieces or 200 g of seafood).**							<0.001
2 or less portions per week.	74.10%	81.40%	76.10%	69.30%	77.60%	71.50%	
3 or more portions per week.	25.90%	18.60%	23.90%	30.70%	22.40%	28.40%	
**How many times per week do you consume industrial bakery (non-home made), like biscuits, puddings, sweets, or cakes?**							<0.001
1 or less portions per week.	72.10%	72.40%	72.20%	54.30%	59.00%	55.60%	
2 or more portions per week.	27.90%	27.60%	27.80%	45.70%	41.00%	44.40%	
**How many times per week do you consume nuts? (portion 30 g).**							<0.001
2 or less portions per week.	70.80%	70.70%	70.80%	57.90%	60.30%	58.60%	
3 or more portions per week.	29.20%	29.30%	29.20%	42.10%	39.70%	41.40%	
**Do you preferably consume chicken, turkey, or rabbit meat instead of beef, pork, hamburgers, or sausages? (portion 100–150 g)**							<0.001
Yes.	82.80%	76.90%	81.20%	79.40%	73.80%	77.80%	
No.	17.20%	23.10%	18.80%	20.60%	26.20%	22.20%	
**How many times per week do you consume cooked vegetables, pasta, rice or other dishes, seasoned with tomato sauce, garlic, onion, or leek slow-cooked with olive oil?**							<0.001
1 or less portions per week.	40.10%	40.30%	40.20%	33.20%	36.90%	34.20%	
2 or more portions per week.	59.90%	59.70%	59.80%	66.80%	63.10%	65.80%	

**Table 4 nutrients-12-02826-t004:** Distribution of Respondents by the most consumed food and beverages.

Most Consumed Food and Beverages.	Before Confinement	During Confinement	*p*
	*n* (%)	*n* (%)	
Alcoholic drinks	11 (1.03)	30 (2.82)	*p* < 0.001
Carbonated beverages	16 (1.50)	15 (1.41)	
Industrial baking	18 (1.69)	30 (2.82)	
Coffee or tea	84 (7.89)	69 (6.48)	
Pork, beef, or lamb	66 (6.20)	70 (6.57)	
Chicken, turkey, or rabbit	218 (20.47)	151 (14.18)	
Cold meat	21 (1.97)	23 (2.16)	
Fruits	99 (9.30)	103 (9.67)	
Nuts	12 (1.13)	13 (2.25)	
Eggs	32 (3.00)	43 (4.04)	
Milk and dairy products	57 (5.35)	49 (4.60)	
Legumes	61 (5.73)	57 (5.35)	
Bread	50 (4.69)	47 (4.41)	
Pasta and/or rice	123 (11.92)	91 (8.54)	
Fish and/or seafood	22 (2.07)	24 (2.25)	
Homemade baking	3 (0.28)	49 (4.60)	
Snacks and sweets	14 (1.31)	34 (3.19)	
Vegetables.	154 (14.46)	156 (14.65)	

**Table 5 nutrients-12-02826-t005:** Physical activity variables.

Variables	Before Confinement			During Confinement		
	Female	Male	Total	Female	Male	Total
**Practice of physical activity weekly.**						
1–3 times per week.	37.50%	29.70%	35.40%	33.40%	29.70%	32.30%
4–5 times per week.	21.90%	43.80%	27.90%	22.20%	27.60%	23.70%
6 or more times per week.	5.20%	15.20%	7.90%	12.90%	19.00%	14.50%
Do not practice any physical activity.	35.40%	11.40%	28.80%	31.50%	23.80%	29.40%
	X^2^ = 108.76; *p* < 0.001			X^2^ = 13.09; *p* = 0.004		
	X^2^ = 39.09 * *p* < 0.001					
**How long do you spend practicing physical activity in each session?**						
10 to 30 min per session.	8.60%	7.20%	8.30%	22.30%	19.00%	21.40%
31 min to 1 h per session.	35.90%	37.60%	36.30%	33.80%	38.60%	35.10%
More than 1 h per session.	20.50%	42.80%	26.60%	12.50%	20.70%	14.70%
Do not practice any physical activity.	30.50%	12.40%	28.80%	31.40%	21.70%	28.70%
	X^2^ = 77.20; *p* < 0.001			X^2^ = 18.88; *p* < 0.001		
	X^2^ = 137.93 *; *p* < 0.001					

* McNemar–Bowker test. X^2^: chi square; *p*: *p*-value.

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
