# Peer review of "Eating Habits and Physical Activity of the Spanish Population during the COVID-19 Pandemic Period"

_nutrients, 2020, doi:10.3390/nu12092826_

Round 1
Reviewer 1 Report
Dear Authors,
you studied an important topic, but the description of research and results needs to be refined. I list my comments below.
- In my opinion the title should be changed because the article does not talk about eating patterns, but about eating habits that include consumption. The title should be rather (or so): Eating habits and physical activity of Spanish population during COVID-19 pandemic period.
- line 21 - is the record (p<0.05) correct?
- In the Introduction, I suggest shortening the first paragraph to one sentence, and writing more about physical activity, because this is what the content is about. If WHO, then World Health Organization is the correct name.
- I suggest not to use the ellipsis.(lines: 45, 65, 179)
- lines 52-53 - Adapt the research objective to the changed title.
- lines 59-62 - The calculation of the sample size is described inaccurately. It is difficult to understand where the sample size comes from.
- In my opinion in Instruments and variables the literature basis for BMI calculation should be provided. Classification of BMI ranges should be described.
- I suggest that You include the questionnaire in the supplementary material.
- line 74 - "food consumption" instead of "consumption food"
- line 106- "and" instead of "or", I think. But I suggest changing the title to Participants characteristics
- line109- "average age" instead of "age average"
- Figure 1 should be put after the Table 2. and should be described in the Results.
- line 114- Change the record as it suggests that BMI was calculated prior to the pandemic.
- Table 1. - I suggest to put the name of the first column as Variables. In BMI category I suggest to change: "low weight" on "underweight"; "Have you gained weight during the confinement?" on "Weight gain during confinement" and remove "yes" in each line; "cooking at home" on "home cooking" and remove "yes".
- In the Results, the description of eating habits should precede the description of physical activity, as shown in the title.
- Table 2. - please tidy up the lines. In the first column I recommend writing "Practice of physical activity weekly" instead of "Do you practice physical activity?" and remove "Yes, between", but stop at "1-3 times" only. And adjust the next variable this way too.
- line 149- "socio-demographic" instead of "socio-demography"
- line 152- I think, should be "There were" instead of "There are". In my opinion, the past tense should be used in all descriptions.
- Table 4.- the title should be in English. I propose the title of this table: Distribution of responses in terms of MD adherence, or the like. I also suggest that you do not include the questions from the questionnaire as the names of the variables. Variable names should be rewritten. The portion size should be included in the methodology.
- line 168 - "Before" instead of "Previously"
- Table 5. -In my opinion the title should be: Distribution of Respondents by the most consumed food and beverages.
- line 183- the value 52.7% is bigger than 29.9%, not smaller.
- line 187- The sentence shoud begin with "Home cooking...."
- line 190- Can you justify the differences in home cooking between men and women? It would be interesting.
- line 212- "sweetened drinks" instead of "sugary drinks"
- line 220- What does it mean "industrial baking" ? Maybe "confectionary" would be better ? Please explain.
- lines 222-223 - the sentence is stylistically incorrect.
- line 226 - Please use the term "physical activity" throughout the article, do not mix with "physical exercises".
- line 229- Zaragoza-Marti is listed in References as a number 16, not 15.
- line 234- I suggest: Regarding weekly session of physical activity...
- line 240- please explain the abbreviation: ICTs.
- In my opinion Conclusions should be redrafted as they should fit the purpose (objective) of the research.
- In References - Dots should be in the abbreviation of journal names. The year of publication should be bolded.
- The English correction is needed.
With Regards
Author Response
Dear reviewer,
Firstly we would like to thank you the time spent on our manuscript, as well as the proposals to improve the quality scientific-technical of it.
After reviewing those proposals, we have proceeded to adapt our manuscript, except some proposals that we think that does not need changes and we should explain why we have make that decision.
The introduction follows a ‘funnel’ pattern, so it goes from the general to the particular. In the first place, allows the reader to understand the particularity of the moment during the study and afterwards we reflect how this situation can impact in the diet and physical activity. Besides, it is reflected how physical inactivity and the consumption of unhealthy food may cause diseases or health problems.
Regarding BMI results, indeed, it was carried out prior to the pandemic. Moreover it was useful and guided us to know the studied population characteristics. Also, we collected loss and gain of weight during confinement, as it is a good predictor of eating and physical activity changes in our population.
The calculation of the sample size has been carried out with a 95% confidence level and a 3% of precision, as the expected proportion of the change is unknown, 0.5 of proportion has been taken. Sample size has been performed of 385 subjects.
BMI calculation, as well as its levels, has not been described as we assume that the reader has knowledge in this field. Furthermore, it would considerably increase the extension of the manuscript and it would make difficult to provide those readers the maximum amount of information with a minimum expression.
Regarding PREDIMED questionnaire, we think it is necessary to add these questions, as viewing the table, better conclusion can be obtained. In addition, it is not necessary to add the used questionnaire, and it would happen the same with the questions about physical activity. In other words, readers should use PREDIMED questionnaire to remind themselves which question correspond to the data reflected in the table. Portions are included and it is also reflected in PREDIMED questionnaire and it is easier for the reader.
Qualitative differences between genders when cooking at home cannot be justified as these variables are not collected and it is an anonymous questionnaire. Although it is not possible to interview the sample to collect new information, we will take account of it for future investigations.
Once more, we really appreciate the time and attention spent in our manuscript. We hope we have met your expectations, with the changes made and the explanation of those which has not been changed.
Kind regards.
Eduardo Sánchez-Sánchez.

Reviewer 2 Report
The concept is interesting and the idea worth pursuing, because such research can be a source of information for a possible second wave of COVID-19 or any other pandemics in the future.
This is a cross-sectional descriptive study based on a self-administered questionnaire was carried out which it evaluated the dietary patterns, consumption and physical activity of the Spanish population before and after confinement period by COVID-19.
Although the dietary data is interesting, it does of the data lacks novelty. To date, quite a few articles have been published about dietary behaviours during the COVID-19, including the Spanish population.
Nevertheless, the Authors did not escape some ambiguities in the text. There are several areas that need to be addressed to help improve the paper.
Please see my specific comments below.
Abstract
It is important to emphasize what novelty this manuscript brings.
Introduction
The introduction can be improved by giving more background of this covering population from other countries in Europe or America. The justification or rational of the novelty for this project needs to be addressed. A lot of these dietary patterns, consumption and physical activity have already been described. Merely selecting a different population of individuals doesn’t seem novel enough.
Authors could consider looking at lifestyle characteristics to improve novelty.
Materials and Methods
Important details on the methods are lacking. Please provide more information on the tool used to assess dietary patterns, selection of participants e.t.c.
Also, please add a literature reference to the questionnaire used (PREDIMED) - e.g.: Schröder et.al, 2011; Martínez-González et.al. 2012?? and add more details about physical activity was assessed - e.g. in supplementary. Perhaps you should include the entire questionnaire used.
Line 91: Method -? I suggest a Data Collection or Procedure.
Please provide more information about anthropometrics data - it was self-reported?
Authors state that it was representative population -
were they adults or all residents of Spain, including children, adolescents, different races?
Maybe it's better to present it in the diagram - (Flow chart of sample collection), including the inclusion/exclusion criteria for the study.
Moreover, why adherence to the Mediterranean Diet was assessed in 2 categories: <9 - low adherences and > 9 high adherences?
Results
Lines 107-109: Please provide a flowchart of the study population - e.g. in Materiał and Methods. Furthermore, it may be worth considering the removal of 4 people (only women) over 71 years old from the data analysis or combining them with the age category of 56-70 years.
Why was there such a detailed division into BMI categories; in the Obesity type III category there are only 9 people, including 8 women and 1 man - what's the point?. Similarly, gained weight during a pandemic > 12 kg?
Please also check the data in Table 2: Do not practice any physical activity (e.g. female - 35.4 vs 30.5%??) and descriptions in the text - lines: 117; 122.
I would also suggest considering the order in which you discuss the results - figure 1? moreover, it is not discussed in the text.
The presentation of results in tables needs to be improved: commas, periods (tab.1); rounding numbers (tab.5); offsets in lines/column (tab.2); Variables/ category instead Questions / answers to questionnaire (tab. 2).
Lines 149-153: Where this data is presented?
Discussion
The discussion, in my opinion, should be strengthened and take into account more publications in this field - concerning the population of equal countries.
Please summarize the most noteworthy findings in the first paragraph and do not describe it later in this chapter (e.g. lines: 209-214).
Please check the meaning of e.g. sentences:
Within our subjects, 52.7% gained weight during confinement, which it is a smaller percentage than reported by Sidor and Rzymski in Poland (29.9%).
Before and during confinement, PREDIMED questionnaire items which presented a higher adherence in our population were: consumption of olive oil for cooking (97.9% vs. 97.6%) - see below line 214- 215 "There was statistic differences within all the items before and during confinement except for olive oil for cooking and vegetables consumption".
Line 229: Our data are closer to the reported by Zaragoza-Martí et al, in 2015 (15) - this is not true - should be (16).
Furthermore, the discussion should be more focused on the results obtained. Consider sorting out the discussion and dividing it into subsections.
References
In my opinion, more new references should be added that should be included in the Introduction and Discussion.
In addition, It is worth considering another wording for the title to reflect the content of the manuscript.
Author Response
Dear reviewer,
Firstly we would like to thank you the time spent on our manuscript, as well as the proposals to improve the quality scientific-technical of it.
After reviewing those proposals, we have proceeded to adapt our manuscript, except some proposals that we think that does not need changes and we should explain why we have make that decision.
Novelty of this research lies on an enlarge number of variables, not only on the knowledge about adherence but answers to each item, and the most consumed food. Moreover, gender differences have been studied. Physical activity research is not just focused on practicing it, also the number of sessions, timing and specially exercises information research. These results may guide public health authorities to create new health policies directed to:
- Advertisements specifically for gender.
- Advertisements specifically for different food, those more and less healthy.
- Increase tax burden on those most consumed food and less healthy.
- Create physical activity programs that use platforms like Youtube for a higher dissemination.
PREDIMED questionnaire reference has been added. Regarding physical activity, the various questions of the questionnaire are shown in the table. We believe that it would be redundant to include the questionnaire again. Besides, tables help us to make the reading more attractive.
Anthropometric parameters, as well as every variable were self-informed, as it is shown in ‘method’ that a self-administered questionnaire disseminated online has been used.
Socio-demography variables shown in table 1, include age and age range, so different group ages can be seen. The study involves the Spanish population, not being relevant in this study the different races, because we do not search for differences between races but for the Spanish population and this criteria must be fulfilled. Inclusion criteria has been added: to be resident in Spain and be ≥ 16 years old.
Two categories has been taken from PREDIMED questionnaire, because are the ones shown in PREDIMED study, the author of this questionnaire.
Categories definition are based on the Spanish Society for the Study of Obesity. Furthermore, pandemic period lasted for more than 60 days and that along with consumption of unhealthy food, physical inactivity and work absence could impact in this weight gain.
Indeed, figure 1 was positioned incorrectly, it has been moved to the corresponding place and its content has been explained.
Regarding the discussion, the most updated publications at present has been used. However we would appreciate if you know any other most recent publication, and guide us in order to extent our knowledge and compare results.
Indeed, olive oil consumption has not any significant statistic differences between periods (p>0.05). The difference was very small and not significant.
Once more, we really appreciate the time and attention spent in our manuscript. We hope we have met your expectations, with the changes made and the explanation of those which has not been changed.
Kind regards.
Eduardo Sánchez-Sánchez.

Reviewer 3 Report
In this study the authors say that Mediterranean diet adherence slightly increased during confinement, additionally, the number of subjects that practiced physical exercises decreased, as well as the time spent weekly, although there was an increase in the number of sessions/week.
There are some important points to review:
- The approval of the ethics committee is lacking
- Because it was not used the Physical Activity (International Physical Activity Questionnaire Short Form (IPAQ-SF) ?
- I don't understand how sample size was calculated
- If adherence to the Mediterranean diet has slightly increased, how can the addition of unhealthy foods be explained?
- Review the tables (g or gr)
Author Response
Dear reviewer,
Firstly we would like to thank you the time spent on our manuscript, as well as the proposals to improve the quality scientific-technical of it.
After reviewing those proposals, we have proceeded to adapt our manuscript, except some proposals that we think that does not need changes and we should explain why we have make that decision.
Activity International Physical Questionnaire Short Form (IPAQ-SF) has not been used because, after consultation with physical activity professionals of our group and other experts, evaluated that it could have some limitations to be used during confinement. For instance, one of the questions in this questionnaire is: ‘During the last 7 days, on how many days did you do moderate physical activities like carrying light loads, bicycling at a regular pace, or doubles tennis?’ These activities cannot be practiced in the majority of the houses during confinement.
The calculation of the sample size has been carried out with a 95% confidence level and a 3% of precision, as the expected proportion of the change is unknown, 0.5 of proportion has been taken. Sample size has been performed of 385 subjects.
Difference between unhealthy food consumption and MD adherence, might be due to the increase of these kind of food by the subjects. However this quantity do not affect to MD adherence. For example, alcohol consumption has increased, also the consumption of 7 glasses of wine. The last one is a positive item for MD adherence.
Once more, we really appreciate the time and attention spent in our manuscript. We hope we have met your expectations, with the changes made and the explanation of those which has not been changed.
Kind Regards.
Eduardo Sánchez-Sánchez.

Round 2
Reviewer 2 Report
Comments and Suggestions for Authors
The article "Eating habits and Physical activity of Spanish population during COVID-19 pandemic period" contributes additional evidence that there are individual differences in eating habits and physical activity of Spanish population in lockdown.
This manuscript has been improved however, a couple more revisions will strengthen the article:
It is still unclear what the novelty of this study is. This should be added in the introduction.
Regarding the discussion, I propose to include the following publication:
- Pellegrini, M.; Ponzo, V.; Rosato, R.; Scumaci, E.; Goitre, I; Benso, A.; Belcastro, S.; Crespi, C.; De Michieli, F.; Ghigo, E.; Broglio, B.; Bo, S. Changes in Weight and Nutritional Habits in Adults with Obesity during the “Lockdown” Period Caused by the COVID-19 Virus Emergency. Nutrients 2020, 12, 2016.
- Górnicka, M.; Drywień, M. E.; Zielinska, M. A. & Hamułka, J. Dietary and Lifestyle Changes During COVID-19 and the Subsequent Lockdowns among Polish Adults: A Cross-Sectional Online Survey PLifeCOVID-19 Study. Nutrients 2020, 12, 2324.
- Balanzá–Martínez, V.; Atienza–Carbonell, B.; Kapczinski, F. & De Boni, R. B. Lifestyle behaviours during the COVID-19 – time to connect. Acta Psychiatr. Scand. 2020, 141, 399–400.
- Daniela Reyes-Olavarría, Pedro Ángel Latorre-Román, Iris Paola Guzmán-Guzmán, Daniel Jerez-Mayorga, Felipe Caamaño-Navarrete, Pedro Delgado-Floody: Positive and Negative Changes in Food Habits, Physical Activity Patterns, and Weight Status during COVID-19 Confinement: Associated Factors in the Chilean Population. Int J Environ Res Public Health 2020 Jul 28;17(15):5431. doi: 10.3390/ijerph17155431.
- De Oliveira Neto, L.; Elsangedy, H.M.; Tavares, V.D.D.O.; Teixeira, C.V.L.S.; Behm, D.G.; Da Silva-Grigoletto, M.E. #TrainingInHome - Home-based training during COVID-19 (SARS-COV2) pandemic: physical exercise and behavior-based approach. Rev. Bras. Fisiol. do Exerc. 2020, 19, 9.
- Federico Scarmozzino, Francesco Visioli. Covid-19 and the Subsequent Lockdown Modified Dietary Habits of Almost Half the Population in an Italian Sample. Foods 2020 May 25;9(5):675. doi: 10.3390/foods9050675.
Furthermore:
Line 117 should be 22.9 should be 23.2 - please check.
Lines 124-125 “Considering gender, the number of unemployed women was higher than men (9.0% vs. 15%)” should be (15% vs. 9%) - please check.
Thank you for your work on this research and this article.
Best regards
Author Response
Dear reviewer,
We are grateful, once again, for your proposals that will improve the quality scientific-technical of our manuscript. Also, thank you for providing us new bibliography to be added in our discussion.
The authors believe that all of your suggestions should be in our manuscript and therefore, we have modified the points you sent us.
Once more, we really appreciate the time and attention spent in our manuscript. We hope we have met your expectations, with the changes made.
Kind regards.
Eduardo Sánchez-Sánchez.
